# Possible Role of GnIH as a Novel Link between Hyperphagia-Induced Obesity-Related Metabolic Derangements and Hypogonadism in Male Mice

**DOI:** 10.3390/ijms23158066

**Published:** 2022-07-22

**Authors:** Rongrong Luo, Lei Chen, Xingxing Song, Xin Zhang, Wenhao Xu, Dongyang Han, Jianyu Zuo, Wen Hu, Yan Shi, Yajie Cao, Runwen Ma, Chengcheng Liu, Changlin Xu, Zixin Li, Xun Li

**Affiliations:** College of Animal Science and Technology, Guangxi University, Nanning 530004, China; luorongrong426@163.com (R.L.); a1045606819@163.com (L.C.); sxx96721@163.com (X.S.); zhangxin9309201@163.com (X.Z.); x8655455@163.com (W.X.); handongyang98@163.com (D.H.); zuojy2009@163.com (J.Z.); 18277127254@163.com (W.H.); sy1065875247@163.com (Y.S.); nct20163031@163.com (Y.C.); marunwen0306@163.com (R.M.); m15565817902@163.com (C.L.); changlin_xu@163.com (C.X.); 18159199585@163.com (Z.L.)

**Keywords:** GnIH, food intake, glucose and lipid metabolism, testis, reproduction

## Abstract

Gonadotropin-inhibitory hormone (GnIH) is a reproductive inhibitor and an endogenous orexigenic neuropeptide that may be involved in energy homeostasis and reproduction. However, whether GnIH is a molecular signal link of metabolism and the reproductive system, and thus, regulates reproductive activity as a function of the energy state, is still unknown. In the present study, we investigated the involvement of GnIH in glycolipid metabolism and reproduction in vivo, and in the coupling between these two processes in the testis level. Our results showed that chronic intraperitoneal injection of GnIH into male mice not only increased food intake and altered meal microstructure but also significantly elevated body mass due to the increased mass of liver and epididymal white adipose tissue (eWAT), despite the loss of testicular weight. Furthermore, chronic intraperitoneal administration of GnIH to male mice resulted in obesity-related glycolipid metabolic derangements, showing hyperlipidemia, hyperglycemia, glucose intolerance, and insulin resistance through changes in the expression of glucose and lipid metabolism-related genes in the pancreas and eWAT, respectively. Interestingly, the expression of GnIH and GPR147 was markedly increased in the testis of mice under conditions of energy imbalance, such as fasting, acute hypoglycemia, and hyperglycemia. In addition, chronic GnIH injection markedly inhibited glucose and lipid metabolism of mice testis while significantly decreasing testosterone synthesis and sperm quality, inducing hypogonadism. These observations indicated that orexigenic GnIH triggers hyperphagia-induced obesity-related metabolic derangements and hypogonadism in male mice, suggesting that GnIH is an emerging candidate for coupling metabolism and fertility by involvement in obesity and metabolic disorder-induced reproductive dysfunction of the testes.

## 1. Introduction

Many studies have confirmed that body energy reserves and metabolic state are relevant modifiers of fertility; forms of metabolic stress ranging from persistent energy insufficiency to morbid obesity are frequently linked to reproductive disorders [1,2,3]. In recent years, there has been increasing evidence of an interaction between metabolic syndrome and testicular function. Metabolic syndrome—obesity in particular—affects testicular function by reducing total testosterone and sex hormone-binding globulin levels, as well as having a detrimental effect on spermatogenesis [4,5]. On the other hand, hypogonadism further increases insulin resistance, which is the main pathophysiological feature of metabolic syndrome [4]. Although the exact pathophysiological mechanisms remain unclear, several neuroendocrine factors including ghrelin, leptin, and kisspeptin appear to play crucial roles in the interaction between metabolic disorder and testicular function [2,6,7,8,9]. Notwithstanding this, our understanding of the neuroendocrine basis for this phenomenon is still incomplete.

In 2000, a novel hypothalamic neuropeptide, gonadotropin inhibitory hormone (GnIH), was the first avian RFamide peptide identified that directly acts on the pituitary gland to inhibit gonadotropin release from the quail hypothalamus [10]. After the discovery of GnIH in birds, GnIH orthologs, known as RFamide-related peptide-3 (RFRP-3), were subsequently identified in a number of mammals [11,12]. After over 20 years of research, it has been established that GnIH plays an important role in regulating mammalian reproduction through the hypothalamic–pituitary–gonadal (HPG) axis via its G protein-coupled receptor 147 (GPR147) [13]. With the development of GnIH/RFRP-3 physiological functions research, it has emerged, as a novel orexigenic neuropeptide participating in appetite regulation, that exogenous administration of GnIH was shown to potently stimulate food intake in chickens and rats [14,15,16]. Usually, increased food intake is closely related to adiposity and abnormal energy metabolism. Anjum et al. determined that GnIH treatments increased fat accumulation accompanied by impaired testicular function through affecting the glucose uptake in adipose tissue and testis of mice [17]. Recently, we demonstrated that intraperitoneally injected GnIH not only triggered food intake accumulation though changes in meal microstructure and caused metabolic syndrome, featuring increased body mass and dyslipidemia, but also affected blood glucose homeostasis caused glucose intolerance, hypoinsulinism, hyperglucagon, and insulin resistance in rats [16]. This evidence for a regulatory role of GnIH in reproduction, appetite, and glucose homeostasis suggests that this neuropeptide may be important for the integration of energy balance and reproductive function. However, whether GnIH is a molecular signal link of energy metabolism and the reproductive system, and thus, regulates reproductive activity as a function of the energy state, is still unknown.

In this context, we hypothesized that GnIH is an emerging candidate for coupling metabolism and fertility, and it may serve as a cellular conduit for relaying information about circulating levels of glucose and lipid to the gonads to regulate fertility. To validate our hypothesis, the effects of chronic intraperitoneally injected GnIH on food intake, meal microstructure, body and tissue mass, and serum biochemical index were first detected in male mice. Subsequently, to investigate the regulating effects of GnIH on glucose and lipid metabolism of chronic intraperitoneally GnIH-injected male mice, fasting blood glucose levels, intraperitoneal glucose tolerance, and insulin tolerance were measured. At the same time, the expression of glucose metabolism-related genes (*Ins*, insulin is a fundamental function of β cells to maintain glucose homeostasis in the body; *Pdx1* and *NeuroD1*, the β cell-specific transcription factors PDX-1 and Neurod1 are critical to maintain β cell function and survival; *Gcg*, glucagon plays a critical role in the regulation of glycemia) and lipid metabolism-related genes (*ACC*, *SCD-1, FASN*, and *LPL*, acetyl-CoA carboxylase catalyzes, stearoyl-CoA desaturase-1, fatty acid synthase, and lipoprotein lipase were critical genes in the process of fat synthesis; *CPT*, carnitine palmitoyltransferase is the rate-limiting enzyme for fatty acid oxidation) were evaluated in the pancreas and eWAT, respectively. Finally, to determine whether GnIH inhibited testicular function was achieved by relaying circulating metabolic signals and/or changing glycolipid metabolism of testis, GnIH and GPR147 mRNA expression was detected in testis of mice under conditions of different energy imbalances. At the same time, we evaluated the expression of glucose (*HK1*, hexokinase 1 is an important enzyme for glucose metabolism; *IR*, insulin receptor gene is integral steps in insulin signaling; *GLUT8*, glucose transporter-8 is a class III sugar transport facilitator predominantly expressed in testis) and lipid (*CD36*, fatty acid transporter 36 is involved in the uptake and transport of fatty acids; *SCD1*, stearoyl CoA desaturase1 gene is involved in the synthesis and regulation of unsaturated fatty acids; *PPAR-γ*, peroxisome proliferator-activated receptor is the main lipogenic transcription factors) metabolism-related genes were evaluated in testis, in addition to sperm count and motility, expression of testosterone synthesis-related genes (*StAR*, steroidogenic acute regulatory protein; *P450scc*, cytochrome p450 family 11 subfamily member; *AR*, androgen receptor), morphological changes, and the correlation of testicular fertility and glycolipid metabolism of testis of GnIH-injected mice.

## 2. Results

### 2.1. Chronic Intraperitoneally Injected GnIH Increases Male Mice Obesity and Photophase Food Intake and Alters Meal Microstructure

The effect of intraperitoneally injected GnIH on the structure of the first meal in male mice fasted for 8 h was initially investigated in our food intake analysis. As shown in Figure 1A, GnIH treatments significantly increased the food intake (*p* < 0.01), duration (*p* < 0.001), and eating rate (*p* < 0.05) of the first meal compared to the vehicle-treated group.

The food intake and meal microstructure were subsequently monitored over 21 days during the photophase and scotophase post-GnIH injection in the ad libitum fed male mice. As shown in Figure 1B, male mice that were intraperitoneally injected with GnIH exhibited significantly increased cumulative food intake during 2–12 h of the scotophase period (*p* < 0.05), whereas there was no difference in the food intake during any period of the photophase and the 1st hour of the scotophase period. However, the values obtained for the meal frequency showed that intraperitoneal injection of GnIH significantly increased the rate of meal frequency during any period of the photophase and scotophase (*p* < 0.05), dramatically increasing during 2–12 h of the scotophase period (*p* < 0.01).

With the increasing food intake, the body weight of GnIH-treated male mice was also significantly increased (*p* < 0.05) (Figure 1C). Furthermore, the male mice injected with GnIH for 21 d resulted in a significant difference in the average daily food intake (*p* < 0.01) and average daily gain (*p* < 0.01) compared to the controls (Figure 1D).

### 2.2. Chronic Intraperitoneally Injected GnIH Alters Organ Indexes and Serum Biochemical Indexes in Male Mice

To investigate the factors that contributed to GnIH-induced body mass being significantly elevated in male mice, the weights of some organs or tissues, as well as the serum biochemical indexes, were measured. As shown in Figure 2A, after 21 d of administration, the weight of testis in male mice intraperitoneally injected with GnIH decreased significantly (*p* < 0.001), whereas the weight of liver and eWAT increased significantly (*p* < 0.05). iWAT and BAT did not change compared with the control group. Furthermore, the values obtained for the serum biochemical parameters showed that intraperitoneally injected GnIH significantly increased the concentrations of LDL-C (*p* < 0.05) and GLU (*p* < 0.01) (Figure 2B). In addition, treatment with GnIH caused no significant change in the serum TG, CHOL, HDL-C, ALT, AST, or the ratio of AST/ALT when compared with control mice.

### 2.3. Chronic Intraperitoneal Injection of GnIH Impaired Glucose Homeostasis in Male Mice

The effect of intraperitoneally injected GnIH on fasting blood glucose levels was first evaluated at different time points in male mice fasted for 8 h. As shown in Figure 3A, the fasting blood glucose levels dramatically increased for 15 min post-GnIH injection (*p* < 0.001) and then gradually decreased from 30 to 135 min after GnIH injection, whereas the blood glucose levels of the vehicle-treated mice showed a slight fluctuation over the 135 min study period. Compared to the vehicle-treated mice, the AUC_fasting blood glucose_ levels were significantly augmented in the male mice injected with GnIH (*p* < 0.01).

In order to study the effect of intraperitoneal injection of GnIH on glucose elimination, intraperitoneal glucose tolerance tests were performed for ad libitum fed mice administered a chronic dose of GnIH. As shown in Figure 3B, a more than 100% increase in the glycemic response was observed 15 min after exogenous glucose administration in all groups. The blood glucose levels peaked at 30 min and then gradually decreased from 15 to 120 min after GnIH injection. In addition, significantly higher glucose levels were detected 15–30 min after the administration of GnIH compared with the control (*p* < 0.05 and *p* < 0.01). The blood glucose concentrations were significantly higher in the GnIH-treated mice than in the vehicle-injected mice, as shown by the area under the curve values (*p* < 0.01).

To determine the effect of intraperitoneally injected GnIH on insulin sensitivity, insulin tolerance was measured in mice administered chronic GnIH. Our results showed that chronic injection of GnIH significantly weakened insulin-induced hypoglycemia between 30 and 75 min after insulin challenge (*p* < 0.05) (Figure 3C). A similar result was observed for the reduced AUC_glucose_ (*p* < 0.01). Taken together, these results indicate that GnIH inhibited exogenous glucose elimination and reduced insulin sensitivity in male mice.

### 2.4. Chronic Intraperitoneal Injection of GnIH Alters the Expression of Genes Related to Glucose and Lipid Metabolism in Pancreas and eWAT

As described above, mice that were chronically administered GnIH exhibited notable glucose intolerance and eWAT accumulation. These data prompted us to further investigate the molecular mechanism underlying GnIH-mediated glucose and lipid metabolism disorder in the pancreas and eWAT of male mice. As shown in Figure 3D, *Ins*, *Pdx1*, and *NeuroD1* mRNA expression levels were markedly decreased (*p* < 0.05), whereas *Glucagon* mRNA expression levels were significantly increased in the pancreas of the mice administered with GnIH (*p* < 0.05). In eWAT, the expression of *ACC* (*p* < 0.01), *SCD1* (*p* < 0.05), *FASN* (*p* < 0.001), and *LPL* (*p* < 0.05) were significantly elevated, accompanied by significantly decreased *CPT* (*p* < 0.05) expression resulting from chronic GnIH treatment.

### 2.5. GnIH and Its Receptor GPR147 Are Involved in Glucolipid Metabolic Disorder-Induced Testicular Dysfunction

To evaluate whether the GnIH/GPR147 system is a molecular signal link metabolism and reproduction system, the expression of GnIH and GPR147 was detected in testis of mice under different energy conditions. Our results showed that *GnIH* and *GPR147* mRNA expression levels were significantly increased in the testis of mice under acute hypoglycemia (*p* < 0.01 and *p* < 0.001) and hyperglycemia (*p* < 0.05 and *p* < 0.001). In addition, 8 h fasting in male mice led to significantly increased *GPR147* mRNA expression (*p* < 0.05), whereas the expression of GnIH was not significantly different between the two groups in testis (Figure 4A).

Subsequently, the effects of chronic intraperitoneal administration of GnIH on glycolipid metabolism and reproductive function of testis in mice were investigated. As shown in Figure 4B,C, GnIH resulted in a significant decrease in the expression of *HK1* (*p* < 0.001), *IR* (*p* < 0.01), *CD36* (*p* < 0.05), and *PPAR-γ* (*p* < 0.01) in testis, which was accompanied by dramatically decreased sperm count and motility levels (*p* < 0.05) and markedly reduced expression levels of *StAR* (*p* < 0.001), *P450scc* (*p* < 0.01), and *AR* (*p* < 0.01) (Figure 4E–G). However, there was no effect on *GLUT8* and *SCD1* expression in testis.

The morphological changes confirmed that GnIH injection altered the testicular cytoarchitecture of mice. Representative examples of histological sections of the testis from the GnIH-treated mice are shown in Figure 4D. In the testis, the control mice showed intact germinal epithelium and all stages of spermatogenesis in seminiferous tubules, as well as an amount of red-stained Leydig cells between seminiferous tubules. Mice treated with GnIH in vivo showed a significant decrease in the number of spermatogenic cells while loosening and even incomplete germinal epithelium in seminiferous tubules. Notably, the number of Leydig cells was significantly decreased in the testis of GnIH-treated mice when compared with that in control mice.

The correlation analysis (Table 1) between testicular reproductive parameters and expressions of testicular glucolipid metabolism-related genes showed that there is a significant positive correlation between the expression of *HK1* and *StAR*: the calculated Pearson’s correlation coefficient is 0.85 (*p* < 0.001) (Table 1). However, there is a significant negative correlation between the expression of *HK1* and *AR* and the expression of *SCD1* and *P450scc*: the calculated Pearson’s correlation coefficient is −0.48 (*p* < 0.001) and −0.68 (*p* < 0.001), respectively. Furthermore, there is a significant positive correlation between the expression of *GLUT 8* and sperm motility as well as the expression of *CD36* and *StAR*: the calculated Pearson’s correlation coefficient is 0.44 (*p* < 0.05) and 0.38 (*p* < 0.05), respectively (Table 1).

## 3. Discussion

Reproduction is closely linked to body energy and metabolic status. The burgeoning field of metabolic reproduction regulation has been gaining momentum due to highly frequent discoveries of new neuroendocrine factors regulating both energy balance and reproduction. Universally throughout the animal kingdom, energy deficits inhibit the reproductive axis, which demonstrates that reproduction is acutely sensitive to fuel availability. GnIH is well known for its crucial role in the regulation of the reproductive axis [12,18,19,20], but in light of accumulating data, it has emerged as a novel orexigenic neuropeptide that affects feeding and is potentially involved in energy metabolism [14,16,21,22]. However, the relationship between GnIH in reproduction and energy metabolism has not been studied. Thus, the aim of this study was to assess whether GnIH is a molecular signal link of metabolism and the reproductive system, and thus, whether it regulates reproductive activity as a function of the energy state in male mice.

The first study to establish that GnIH, as a novel orexigenic peptide, stimulated food intake was reported by Tachibana et al. in chicks [14]. Subsequently, McConn et al. confirmed that GnIH promoted not only feeding but also high expression in fasting chicks via intracerebroventricular (ICV) injection of GnIH, indicating that it is an innate hunger factor. In addition to chicks [23], a similar result that GnIH elevated cumulative food intake was also observed in sheep, mice, and cynomolgus monkeys via ICV infusion [24]. Our most recent previous study first confirmed that the effect of acute and chronic intraperitoneally injected GnIH on food intake was similar to that of ICV injection in rats [16]. The present study validated the effect of peripherally treated GnIH on appetite in both fasting and ad libitum fed mice. Notably, mice that were intraperitoneally injected with GnIH exhibited significantly increased food intake during the 2 to 12 h period of the scotophase, whereas no change was observed during the photophase or the first hour of the scotophase post-GnIH injection in ad libitum fed mice. Even though these findings suggested that exogenous GnIH could not alter the natural preference of mice for nocturnal food ingestion, they showed a discrepancy in rats, because GnIH significantly increased food intake during the photophase. In rodents, reproductive status is primarily driven by day length (photoperiod). Photoperiodic mammals rely on the annual cycle of changes in nocturnal secretion of a pineal hormone, melatonin, to drive their reproductive responses. Several studies have confirmed that photoperiod and melatonin regulate the synthesis and release of GnIH involved in many seasonal breeding species. The expressions of GnIH precursor mRNA and GnIH-ir fiber density decreased under a short-day photoperiod compared to a long-day photoperiod in Syrian hamsters [25] and Soay sheep [26]. In contrast to findings in rodents and Soay sheep, GnIH is increased during the non-breeding season in Blackface sheep [27]. In quail, the expression of GnIH was photoperiodically controlled and increased under short-day photoperiods, when the duration of melatonin secretion increases [28]. However, the interrelation between melatonin and GnIH seems uncertain. Melatonin induces the expression of GnIH in quail, whereas melatonin inhibits the expression of GnIH in hamsters [29]. Together, these findings suggest that the relationship between light perception and GnIH is more complicated than our investigation suggests, and this issue needs further study in the future. Based on the above findings, a possible explanation for the discrepancy is that light cycle and even melatonin are involved in this process, and thus, result in different effects on GnIH-induced food intake accumulation in different species. The analysis of the meal pattern is of primary importance to assess the mechanisms regulating feeding behavior. Interestingly, we observed that GnIH markedly elevated meal duration and eating rate of the first meal in fasting mice, while significantly increasing meal frequency during the second hour and 3 to 12 h period of the photophase and the whole period of the scotophase in ad libitum fed mice, consistent with the results of the meal microstructure analysis in intraperitoneally GnIH-treated rats [16]. These data corroborate the previous suggestion that intraperitoneally injected GnIH increased food intake by causing changes in meal microstructure, as influenced by the light cycle.

Abnormal food intake and changes in meal microstructure very easily cause fat accumulation and body obesity [30,31,32]. Our previous study determined that intraperitoneally injected GnIH not only increased food intake and altered meal microstructure but also increased obesity in rats [16]. Similar results were found in our present study: body mass was gradually but notably increased from 8 to 21 d in the GnIH-treated ad libitum fed mice, exhibiting markedly increased average daily food intake and average daily gain over 21 days. The results of the tissue and organ mass investigation showed that eWAT accumulated and liver gains increased, resulting in increased body weight in the chronic intraperitoneally GnIH-treated mice. Our results correspond with the previous study in that the mass of liver was increased by the Chronic Intracerebroventricular Infusion of GnIH in lean male C57BL/6J mice, despite the mass of white adipose tissue being unchanged [33]. Moreover, the concentrations of serum LDL-C and glucose were notably increased in the GnIH-treated mice, which was supported by and corresponded with the results of our food intake and body and organ mass measurements. Another previous study performed in mice also indicated that GnIH mediated increased uptake of nutrients (glucose and TGs) in the adipose tissue, resulting in the accumulation of fat [17]. Our findings are in agreement with previous studies, suggesting that GnIH-triggered food intake accumulation causes metabolic syndrome, featuring increased body mass, hyperglycemia, and dyslipidemia.

Based on the above research, we were prompted to investigate the effects of GnIH on glucose and lipid metabolism in mice. Our results are consistent with our previous research on rats, suggesting that intraperitoneally injected GnIH could instantly and markedly elevate blood glucose levels within 15 min, which lasted over 135 min in fasting mice. Similar results were also observed in an intraperitoneal glucose tolerance test that showed that the blood glucose levels were significantly increased after exogenous glucose administration in mice administered with GnIH. These results indicated that GnIH is involved in glucose metabolism, suggesting that intraperitoneally injected GnIH not only induced hyperglycemia but also blunted the sensitivity of mice to exogenous glucose administration and decreased the glucose elimination response. In addition, the results of insulin tolerance tests showed that chronic intraperitoneal injection of GnIH significantly weakened insulin-induced hypoglycemia after insulin challenge over 120 min. Equal results of insulin tolerance were found in GnIH-treated rats in our previous study. These data corroborate the previous suggestion that chronic intraperitoneally administered GnIH exhibited notable glucose intolerance with simultaneous low insulin sensitivity. Even though we have revealed that decreased uptake of glucose, suppression of insulin signaling, and increased inflammatory response may be responsible for GnIH-induced insulin resistance in rats, the precise mechanism of GnIH-induced insulin resistance in different species is complicated and worth further research.

We also attempted to decipher the molecular mechanism underlying the GnIH-mediated hyperglycemia and dyslipidemia in glucolipid-regulating organs. We observed that intraperitoneally injected GnIH could significantly reduce the gene expression of insulin and insulin promoter factors, whereas it increased glucagon gene-expression levels in pancreas of mice. These results explained the possible reasons for GnIH-induced hyperglycemia in mice due to decreased insulin synthesis accompanied with increased glucagon synthesis. In our previous investigations, the immunofluorescence doublestaining data consistently revealed that GnIH was primarily colocalized with glucagon, whereas GPR147 primarily colocalized with insulin in pancreatic islets of rats [16]. A related study performed using mouse pancreatic islets and cells from alpha TC1 clone 6 showed that GnIH promotes the survival of alpha cells via GPR147 [34]. Moreover, GnIH dramatically promoted pancreatic islet hyperplasia with the simultaneous synthesis and secretion of insulin being significantly decreased; however, glucagon had the opposite effect in both mice and rats. The above results suggest that GnIH functions in an autocrine and/or paracrine manner in the pancreas to regulate glucose homeostasis. In addition, the present study first revealed that GnIH affects lipid metabolism of eWAT through elevated expression of the critical limiting enzyme in fatty acid biosynthesis accompanied with inhibited fatty acids oxidation. These may be the reasons why GnIH chronic treatment triggered increased body mass and dyslipidemia in mice. Our findings are supported by a previous study, which demonstrated that GnIH induced the increased expression of GLUT4 together with increased uptake of TGs in the adipose tissue of male mice [17]. It is well known that insulin and glucagon play important but opposite roles in glucose and lipid homeostasis. Insulin increased glucose utilization and expression of multiple genes encoding de novo lipogenesis and cholesterol biosynthesis enzymes, while glucagon showed the opposite effects in WAT [35,36]. These insulin and glucagon responses in WAT make our results seem self-contradictory, that is, the GnIH-induced insulin decrease and glucagon increase seem unable to trigger eWAT, liver, and accumulated body weight. However, our previous study confirmed that GPR147 expressed in WAT, liver, and skeletal muscle in rats, suggesting that GnIH can directly modulate the physiological function of these tissues and may change lipid metabolism accordingly. Simultaneously, we have confirmed that GnIH increased insulin resistance in WAT and liver [16]. Evidence indicated that insulin resistance, especially adipose tissue insulin resistance as a pathogenic factor, is associated with the development of obesity, Type 2 diabetes, hypertension, and cardiovascular disease, but the mechanistic significance of this concept is not fully understood [37]. Based on these findings, we suggested that the fact that GnIH induced increased body mass was a result that was balanced by many influencing factors. Apart from the influence of insulin and glucagon on lipid metabolism, the direct effects of GnIH on lipometabolic organs or tissues, food intake, insulin resistance, etc. may contribute to accumulated body mass in mice. On the basis of the present study and previous studies, GnIH appears to play an important role in glucose and lipid metabolism. Further studies are required to determine whether GnIH is involved in the development of obesity, Type 2 diabetes, hypertension, and cardiovascular disease in different disease model animals and humans.

Since gonadotropin-inhibitory hormone (GnIH) was first identified in Japanese quail as an inhibitor of gonadotropin synthesis and release, there are several studies that have confirmed that GnIH, acting via GPR147, can suppress testosterone secretion and spermatogenesis by acting at the level of the testis from birds to mammals [38,39,40,41,42]. Anjum et al. found that GnIH treatment in vivo not only showed dose-dependent regressive changes in the histological features, but also showed a significant decrease in the serum testosterone level and expressions of steroidogenic factors in a dose-dependent manner when compared with the control [17,43]. Our findings are in agreement with previous studies, suggesting that GnIH plays an inhibitory role in testicular activities. It is worth noting that fertility is gated by nutrition and the availability of stored energy reserves, but the cellular and molecular mechanisms that link energy stores and reproduction are not well understood [2]. Several studies and our previous research have confirmed that GnIH can exert direct or indirect effects on feeding and metabolism, as well as on multiple components of the hypothalamic–pituitary–gonadal (HPG) axis [20,21]. This evidence prompted us to hypothesize that GnIH may serve as a molecular motif integrating the control of metabolism and reproduction. Interestingly, we were agreeably surprised to observe that the expression of GnIH and GPR147 were markedly increased in the testis of mice under conditions of energy imbalance, such as fasting, acute hypoglycemia, and hyperglycemia. In addition, qRT-PCR results showed that chronic intraperitoneally injected GnIH could dramatically inhibit glycolysis and fatty acid intake, as well as decrease the insulin sensitivity of testis in mice. Previous studies have confirmed that treatment with GnIH both in vivo and in vitro showed decreased uptake of glucose by downregulation of *GLUT 8* and *IR* expressions in the testis [17]. Our data corroborate the previous suggestion that GnIH/GPR147 system could direct perception of metabolic information and pass this to the testis while simultaneously regulating testicular energy metabolism by causing changes in disruption of insulin action and glucolipid homeostasis. Evidence has determined that testicular glucose and lipid metabolism influenced steroidogenesis and spermatogenesis, consequently contributing to male infertility [44,45,46,47]. A previous study showed that GnIH treatment in vivo caused downregulation of GLUT8 and IR expression, resulting in the decreased uptake of glucose, which in turn resulted in the decreased synthesis of testosterone and testicular activity [17]. Our results showed that there is a correlation between testicular reproductive parameters and expressions of testicular glucolipid metabolism-related genes, which was supported by and corresponded with previous results, suggesting that GnIH-induced decreased uptake of glucose and fatty acid may be directly responsible for the inhibition of testicular steroidogenesis, spermatogenesis, and testicular activity.

In summary, the results of this study indicate that GnIH causes food intake accumulation, metabolic syndrome, and weight gain through changes in dietary microstructure in male mice. In addition, we elucidated the molecular mechanism underlying GnIH-mediated hyperglycemia and dyslipidemia: GnIH disturbed the synthesis of insulin and glucagon in the pancreas as well as fatty acid biosynthesis and fatty acid oxidation in eWAT. In addition to GnIH-triggered hyperphagia-induced obesity-related metabolic derangements on a whole-body level, GnIH could also direct perception of metabolic information in the testis level and induce testicular energy metabolism disorder, resulting in hypogonadism in male mice. On the basis of the present study, GnIH apparently does not affect feeding, but in light of accumulating data, it has emerged as one of the major conduits in relaying body metabolic status information to testis and accordingly modulating the reproductive strategies to be adopted by animals, which suggests GnIH-GPR147 signaling as a possible pathway providing a link between metabolism and reproduction. Nowadays, the growing prevalence of metabolic disorders, including morbid obesity, diabetes, and anorexia, has become one of the major threats to human reproductive health. In this context, further studies are required to establish whether the GnIH is associated with hypogonadism, as observed during these metabolic disorders. Understanding the physiological mechanisms of GnIH connecting metabolism and reproduction could improve ways of treating or preventing pathological conditions associated with these crucial phenomena.

## 4. Materials and Methods

### 4.1. Animals and GnIH

All of the experiments were performed in accordance with the guidelines of the regional Animal Ethics Committee and were approved by the Guangxi University Ethical Committee (Project ID GXU2020-069). Specific pathogen-free male Kunming mice (6 weeks old, 20 ± 2 g) were obtained from the inbred colony maintained in the Guangxi Medical University of Laboratory Animals Center. The animals were maintained under constant conditions of light (12:12 light–dark cycle) and temperature (25 °C) with free access to rodent food (65% carbohydrate, 25% protein, and 10% fat) and water.

A portion of 30 male mice (*n* = 30, 15 per group) received chronic intraperitoneal injections of GnIH (20 μg/100 μL, total 200 μL of GnIH dissolved in a 0.9% saline solution, the dose of GnIH used in the present study was based on our preliminary experiments and previous studies [16,43]) twice a day (7 a.m. and 7 p.m.) for 21 days. Mice in the control group received vehicle only.

Another portion of male mice (*n* = 60, 15 per group) were fasted for 8 h and then used to induce different energy imbalance mice models. Acute hyperglycemia was induced by an intraperitoneal injection of glucose (2.2 g/kg), as previously shown [48]. Acute hypoglycemic mice were administered intraperitoneal insulin to induce a blood glucose nadir of 50 mg/dL at 0.75 h after injection as previously described [49]. Blood glucose was immediately measured using a blood glucose meter (FreeStyle OptiumNeo, Abbott Laboratories, Shanghai, China) from tail vein blood samples at 20 min after an injection of glucose or insulin. Fasted mice were fasted for 8 h, and ad libitum fed mice were allowed water and food ad libitum.

GnIH (Catalog No. 048-52, Phoenix Pharmaceuticals Co.,Ltd., Burlingame, CA, USA) was used in the present study. The peptide was stored in powder form at −80 °C, weighed, and dissolved in vehicle immediately before administration.

### 4.2. Food Intake, Meal Microstructure, Weight, and Serum Biochemical Index Measurements

All male mice (*n* = 30, 15 per group) used in food intake analyses were housed individually. The body mass of chronic intraperitoneal GnIH or vehicle injected male mice was measured every day (7 a.m.) for 21 days, while food intake was determined by placing preweighed standard food pellets in cages and weighing the remained food after GnIH injection 24 h in individual cages every day (7 a.m.) for 21 days. The structure of first meal in the fasting mice and meal microstructure monitoring was conducted as detailed in a previous study [50]. After 21 days treatment, the mice were sacrificed by decapitation 15 min later after the last injection. Subsequently, serums were collected to measure total triglyceride (TG, Catalog No. U82885040, Uritest Co., Ltd., Guilin, China), low-density lipoprotein cholesterin (LDL-C, Catalog No. U83085045, Uritest Co., Ltd., Guilin, China), high-density lipoprotein cholesterol (HDL-C, Catalog No. U83085045, Uritest Co., Ltd., Guilin, China), aspartate aminotransferase (AST, Catalog No. U83085045, Uritest Co., Ltd., Guilin, China), alanine aminotransferase (ALT, Catalog No. U80785040, Uritest, Guilin, China), glucose (GLU, Catalog No. U83781050, Uritest Co., Ltd., Guilin, China), and cholesterol (CHOL, Catalog No. U82785040, Uritest Co., Ltd., Guilin, China) levels using an automatic biochemical analyzer (URIT-8021AVeT, Uritest Co., Ltd., Guilin, China). In addition, testis, brown adipose tissue (BAT), epididymal fat white adipose tissue (eWAT), and inguinal white adipose tissue (iWAT) were excised and collected as described in previous studies [51], and then weighed. The organ index was calculated as the weight of different organs divided by the body weight.

### 4.3. Blood Glucose Measurements

For male mice (*n* = 30, 15 per group) that had fasted for 8 h or received chronic intraperitoneal injections of GnIH or vehicle as described above, blood glucose was immediately measured using a blood glucose meter (FreeStyle Optium Neo, Abbott Laboratories, Shanghai, China) from tail vein blood samples at specific time points (15 min prior to and 0–120 min after an injection of GnIH).

### 4.4. Glucose Tolerance Test

Before the glucose tolerance test, chronic GnIH-injected mice received intraperitoneal injections of GnIH twice a day for 21 days as described above. After fasting for 8 h, mice (*n* = 30, 15 per group) were all intraperitoneally injected with glucose (Kelun Medicine Co., Ltd., Hubei, China) at a dose of 2 g/kg body weight. Then, 15 min later, GnIH was intraperitoneally injected into the mice of the GnIH-treatment and control groups. Subsequently, blood glucose was immediately measured from tail vein blood samples at specific time points as described above. The mice of the control group received saline instead.

### 4.5. Insulin Tolerance Test

Before the insulin tolerance test, chronic GnIH-injected mice received intraperitoneal injections of different doses of GnIH twice a day for 21 days as described above. After fasting for 8 h, mice (*n* = 30, 15 per group) in the control and chronic treatment groups were all intraperitoneally injected with insulin (Wanbang Co., Ltd., Jiangsu, China) at a dose of 0.5 IU/kg body weight. Then, 15 min later, GnIH was intraperitoneally injected into the mice of all groups. Subsequently, blood glucose was immediately measured from tail vein blood samples at specific time points, as described above.

### 4.6. Gene Expression

The pancreas, eWAT, and testis from the control mice or mice having undergone chronic GnIH intraperitoneal administration for 21 days were excised. In addition, the testis of mice (*n* = 60, 15 per group) under ad libitum, fasted, hyperglycemia, and acute hypoglycemia conditions were also collected. Relative real-time RT-PCR was performed as described in our previous study [52]. Amplification reactions were conducted in triplicate using gene-specific primers designed based on the clone sequences shown in Table 2. The quantitative RT-PCR was performed using the 2^−^^ΔΔCT^ method with GAPDH as the internal control for normalization.

### 4.7. Sperm Parameter Analysis of Epididymis

After 21 days of treatment, chronic GnIH-injected and control mice (*n* = 30, 15 per group) were anesthetized. The epididymides were harvested and cut into fragments with a length of about 2 mm with ophthalmic scissors as far as possible, then placed in a petri dish containing HEPES buffer solution, and the tissue was repeatedly squeezed with ophthalmic tweezers to free the sperm in the liquid. The liquid was collected into EP tubes and then transferred to an incubator for incubation for 10 min. The semen sample was collected with the sampler and added to the counting chamber (15 μL on each side) of the blood cell counting board to calculate the sperm density. Semen samples from each mouse were analyzed three times and averaged. The morphology of sperm was observed under light microscope (Nikon) by conventional methods.

### 4.8. Histological Examination of Testicular Tissue

The testicular tissue from the control mice or mice having undergone chronic GnIH intraperitoneal administration for 21 days (*n* = 30, 15 per group) were excised and fixed in neutral-buffered formalin. Tissue samples were dehydrated in a graded ethanol series and embedded in paraffin wax followed by histological sectioning (5 μm). Paraffin-embedded testicular tissues were stained with hematoxylin and eosin (H & E). For histological study of the testes, five sections from each testis and two sections from each slide were randomly selected and observed under a light microscope (Nikon Corporation, Tokyo, Japan). In each section, the number of spermatogenic cells and Leydig cells were counted according to the method in previous studies [43,53,54]. The photomicrographs were captured by using a calibrated photomicroscope connected to a computer equipped with AxioVision 4.8 software (Version 4.8, ZEISS, Berlin, Germany).

### 4.9. Statistical Analyses

Numerical data are expressed as the mean ± S.E.M. The statistical analysis was evaluated by unpaired two-tailed Student’s *t*-test or one-way ANOVA followed by a post hoc Tukey’s test. Significant correlations between testicular reproduction parameters and expressions of testicular glucolipid metabolism-related genes were searched for with a Pearson product-moment correlation analysis.

SPSS software (Version 13.0, SPSS Inc., Chicago, IL, USA) was used for statistical analyses, and *p* < 0.05 was considered to indicate a significant difference.

## Figures and Tables

**Figure 1 ijms-23-08066-f001:**
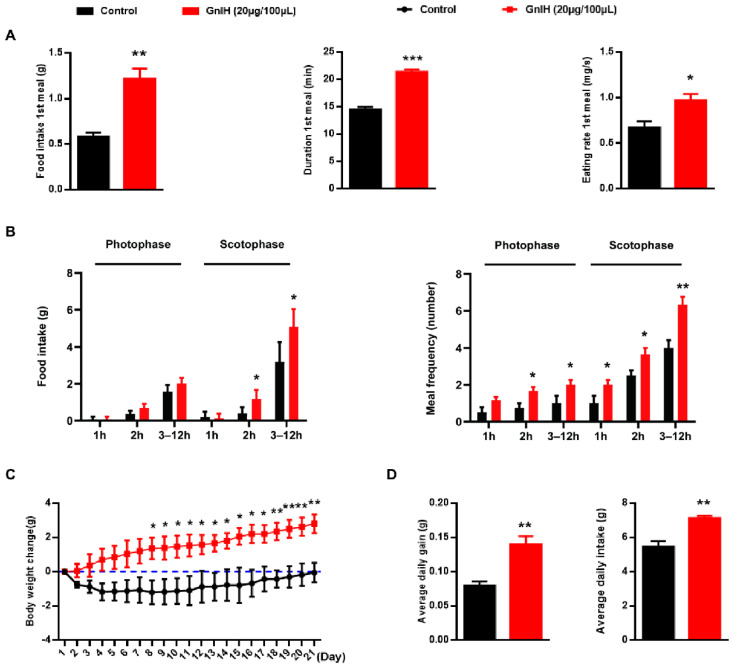
Chronic intraperitoneally injected GnIH increases male mice obesity and photophase food intake and alters meal microstructure. (**A**) Effect of GnIH injected intraperitoneally on the structure of first meal in the fasting mice. (**B**) The food intake and meal frequency were monitored over 21 d during different periods of the photophase and scotophase post-GnIH or vehicle injection in the ad libitum fed mice. (**C**) The body weight change. (**D**) Average daily gain and average daily intake of mice intraperitoneally injected with GnIH or vehicle for 14 d. *n* = 15/group. The data are presented as the mean ± SEM. * *p* < 0.05, ** *p* < 0.01, and *** *p* < 0.001 GnIH 20 μg/100 μL vs. vehicle.

**Figure 2 ijms-23-08066-f002:**
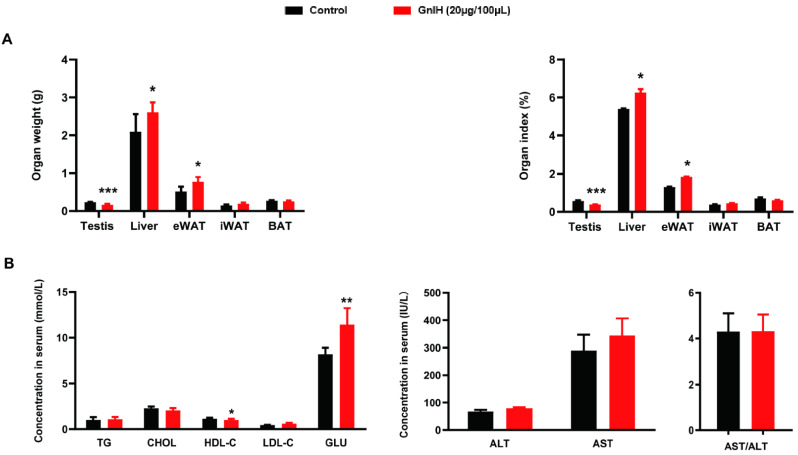
Chronic intraperitoneally injected GnIH alters organ indexes and serum biochemical indexes in male mice. (**A**) The organ weight and organ index in male mice intraperitoneally injected with GnIH or vehicle for 21 d. (**B**) The concentration of TG (total triglyceride), GLU (glucose), CHOL (cholesterol), HDL-C (high-density lipoprotein cholesterol), LDL-C (low-density lipoprotein cholesterin), ALT (alanine aminotransferase), and AST (aspartate aminotransferase) in the serum of chronic GnIH or vehicle-treated male mice. *n* = 15/group. The data are presented as the mean ± SEM. * *p* < 0.05, ** *p* < 0.01, and *** *p* < 0.001 GnIH 20 μg/100 μL vs. vehicle.

**Figure 3 ijms-23-08066-f003:**
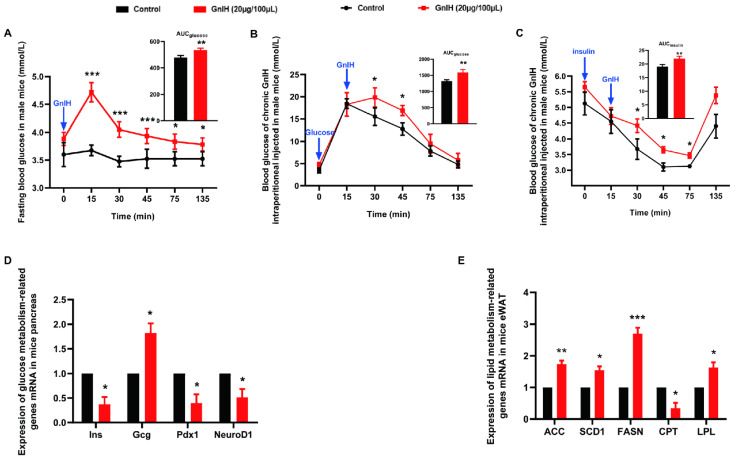
Chronic intraperitoneal injection of GnIH impaired glucose homeostasis in male mice. (**A**) The fasting blood glucose levels were measured at different time points after intraperitoneally injecting GnIH or vehicle into mice that had fasted for 8 h. The upper panel shows the total area under the curve (AUC) for fasting blood glucose after GnIH or vehicle injection from 0 to 120 min. (**B**) For the intraperitoneal glucose tolerance test, blood glucose concentrations were measured in ad libitum fed mice that had been administered with GnIH or vehicle. The upper panel shows the total AUC values for blood glucose or insulin after the administration of GnIH or vehicle from 0 to 120 min; (**C**) The blood glucose levels during the insulin tolerance test were measured in mice injected intraperitoneally with GnIH or vehicle. The panel shows the total AUC for blood glucose after the administration of different doses of GnIH or vehicle from 0 to 120 min. (**D**,**E**) The mRNA expression of glucose and lipid metabolism-related genes in the pancreas and eWAT of mice administered with GnIH or vehicle for 21 d, respectively. *n* = 15/group. The data are presented as the mean ± SEM. * *p* < 0.05, ** *p* < 0.01, and *** *p* < 0.001 GnIH 20 μg/100 μL vs. vehicle.

**Figure 4 ijms-23-08066-f004:**
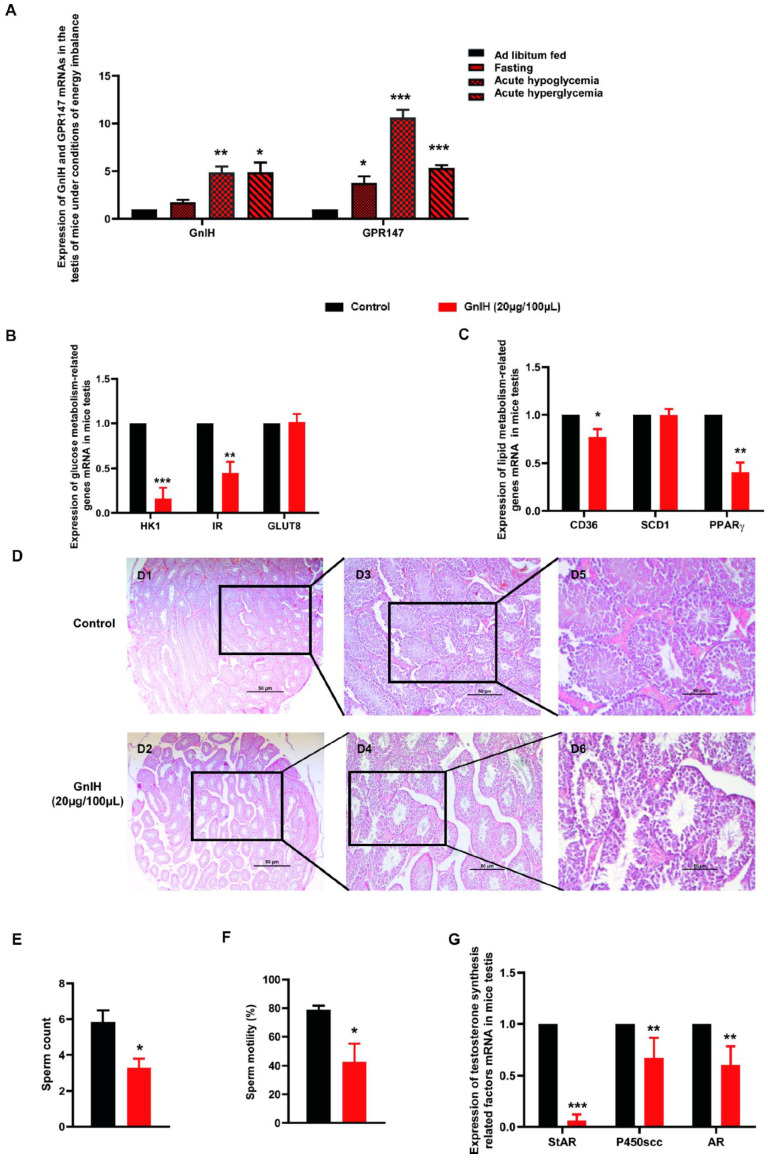
GnIH and its receptor GPR147 are involved in glucolipid metabolic disorder-induced testicular dysfunction. (**A**) The mRNA expression of GnIH and its receptor GPR147 in the testis of mice under conditions of energy imbalance. (**B**,**C**) The mRNA expression of glucose and lipid metabolism-related genes in testis of mice receiving chronic intraperitoneal injection of GnIH or vehicle for 21 d. (**D**) The morphological changes in testis of mice receiving chronic intraperitoneal injection of GnIH (D1, D3 and D5) or vehicle (D2, D4 and D6) for 21 d. D5 and D6 was magnified from D3 and D4 while D3 and D4 was magnified from D1 and D2, respectively. Scale bar = 50 μm. (**E**,**F**) The sperm count and sperm motility of epididymis in mice receiving chronic intraperitoneal injection of GnIH or vehicle for 21 d. (**G**) The mRNA expression of testosterone synthesis-related genes in mice receiving chronic intraperitoneal injection of GnIH or vehicle for 21 d. *n* = 15/group. The data are presented as the mean ± SEM. * *p* < 0.05, ** *p* < 0.01 and *** *p* < 0.001 GnIH 20 μg/100 μL vs. vehicle.

**Table 1 ijms-23-08066-t001:** The correlation analysis between testicular reproductive parameters and expressions of testicular glucose metabolism-related genes.

	*HK1*	*IR*	*GLUT8*
*StAR*	r = 0.85 ** *p* < 0.001	NS	NS
*P450scc*	r = 0.08 *p* > 0.05	NS	NS
*AR*	r = −0.48 ** *p* < 0.001	NS	NS
Testis index	r = 0.25 *p* > 0.05	r = −0.07 *p* > 0.05	r = −0.18 *p* > 0.05
Sperm motility	NS	r = −0.06 *p* > 0.05	r = 0.44 * *p* < 0.05
	*CD36*	*SCD1*	*PPARγ*
*StAR*	r = 0.38 * *p* < 0.05	r = 0.35 *p* > 0.05	r = 0.35 *p* > 0.05
*P450scc*	r = −0.30 *p* > 0.05	r = −0.68 ** *p* < 0.001	r = −0.13 *p* > 0.05
*AR*	r = 0.32 *p* > 0.05	r = 0.27 *p* > 0.05	r = 0.28 *p* > 0.05
Testis index	r = −0.12 *p* > 0.05	r = −0.17 *p* > 0.05	r = 0.09 *p* > 0.05
Sperm motility	NS	NS	r = −0.09 *p* > 0.05

The data are presented as the mean ± SEM. * *p* < 0.05 and ** *p* < 0.01, NS, not significant, GnIH 20 μg/100 μL vs. vehicle. The testis index in each mouse was calculated as the weight of testis divided by the body weight.

**Table 2 ijms-23-08066-t002:** Primers and annealing temperature for relative real-time RT-PCR.

Genes	Primer Sequence (5′–3′)	Annealing (°C)	Gene Accession Number
*StAR*	F: AAGGAAAGCCAGCAGGAGAAC	60	XM_021169998.1
	R: TCCATGCGGTCCACAAGTT		
*P450scc*	F: CAGATGCAGAGTTTCCAA	60	NM_019779.4
	R: TGAGAAGAGTATCGACGCATCCT		
*AR*	F: CTGGGAAGGGTCTACCCAC	60	XM_021188305.2
	R: GGTGCTATGTTAGCGGCCTC		
*HK1*	F: TGCCATGCGGCTCTCTGATG	60	XM_006513247.2
	R: CTTGACGGAGGCCGTTGGGTT		
*GLUT8*	F: TTCATGGCCTTTCTAGTGACC	60	XM_036162406.1
	R: GAGTCCTGCCTTTAGTCTCAG		
*IR*	F: GCAGTTTGTGGAACGGTGCT	55	NM_001330056.1
	R: CCAGGCACTCTTTGTGGCAG		
*CD36*	F: CAGATGACGTGGCAAAGAAC	55	NM_001159558.1
	R: TGGCTCCATTGGGCTGTA		
*PPARγ*	F: TCACAAGAGCTGACCCAATGGT	55	NM_001308354
	R: ATAATAAGGTGGAGATGCAGGTTCTAC		
*SCD1*	F: AGGCCTGTACGGGATCATACT	60	NM_009127.4
	R: AGAGGCTGGTCATGTAGTAG		
*Gcg*	F: CCACTCACAGGGCACATTC	55	XM_006516200.5
	R: CGGTTCCTCTTGGTGTCA		
*Pdx1*	F: CCCCAGTTTACAAGCTCGCT	55	NM_008814.4
	R: CTCGGTTCCATTCGGGAAAGG		
*NeuroD1*	F: CTTGGCCAAGAACTACATCTGG	55	XM_021193023.2
	R: GGAGTAGGGATGCACCGGGAA		
*Ins*	F: GCTTCTTCTACACACCCATGTC	55	XM_021204833.1
	R: AGCACTGATCTACAATGCCAC		
*GnIH*	F: GAGGAATCCCAAAAGGGGTAAAGG	60	XM_021165817.2
	R: GTGATGCGTCTGGCTGTTGTTCT		
*GPR147*	F: AGCCTCACCTTCTCCTCCTACTACC	60	XM_021205580.2
	R: AGTGATAAGGTTGTCCACAAGGGTT		
	R: TCCACCACCCTGTTGCTGTA		

## Data Availability

The data presented in this study are available on request from the corresponding author.

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
