# Peer review of "Possible Role of GnIH as a Novel Link between Hyperphagia-Induced Obesity-Related Metabolic Derangements and Hypogonadism in Male Mice"

_ijms, 2022, doi:10.3390/ijms23158066_

Round 1
Reviewer 1 Report
The Manuscript: Possible Role of GnIH as a Novel Link Between Hyperphagia- Induced Obesity-Related Metabolic Derangements and Hypogonadism in Male Mice, by Luo et al, explores the effects of intraperitoneal GnIH injections in male mice.
Major comments are as follows:
In order to consider mice hypogonadal, testosterone (and LH) measurements are necessary. Did the weight of seminal vesicles change due to GnIH treatment?
It is not clear how the authors concluded: “Mice treated with GnIH in vivo showed a significant decrease in the number of spermatogenic cells while loosening and even incomplete germinal epithelium in seminiferous tubules. Notably, the numbers of testicular interstitial cells were significantly decreased in the testis of GnIH-treated mice when compared with control mice”. In general, histology analysis is not very convincing. Testis tissue is hard to process without artifacts that may look like pathology. Without more detailed analyses and other techniques (immunohistochemistry), one cannot conclude on a number of interstitial cells or the number of spermatogenic cells. In relation to this, presented results imply that spermatogenesis is obviously affected, but not abolished. It would be interesting to see if these changes are reversible i.e. what happens after cessation of GnIH administration.
A previous very similar study with similar outcomes, by Anjum et al, used the highest dose of 2 ug GnIH. How did the authors decide on a 10 times higher dose? Please state clearly if the daily dose was 20 or 40 ug/day.
Clearly state how the animals were housed for food intake analyses (individually or in groups).
The number of animals used for each analysis must be clearly stated in subsections of material and methods and/or in the figure legends.
Minor comments
Revise Y-axis titles in Fig 2, Fig 3, Fig 4...
Please specify the meaning of “testis index” in table 2.
English language and grammar must be corrected throughout the Manuscript.
Results obtained by Anjum et al 2016, should be better discussed and put in the context of the present research.
Reviewer 2 Report
The authors investigate gonadotropin-inhibitory hormone as a link between hyperphagia induced metabolic syndrome and hypogonadism.
Comments:
Results:
Line 116-I am not sure it is surprising that an increase in food leads to weight gain. Should remove the word interesting
Figure 4-was the investigator blinded to the treatment for the histological examination?
Discussion:
Line 281-This statement needs further discussion and a link to previous research- Possible explanation is that light cycle has different effects on GnIH-induced food intake accumulation in different species.
Methods:
It is not clear in the methods what the controls were. ie were they injected with the vehicle?
Why was this amount of GnIH used? Previous publication to justify concentration?
Was the serum levels of leptin quantified: doi: 10.4103/aja.aja_98_18
or gherlin and kisspeptin? These were highlighted in the introduction
Reviewer 3 Report
In this manuscript the effect of gonadotropin-inhibitory hormone (GnIH) administration on glucose and lipid metabolism, body weight as well as reproduction in male mice was investigated. In my opinion, manuscript is interesting and well-written. Nevertheless, I have some comments to the Authors:
1. Introduction: L 77: In my opinion “expression of glucose” should be replaced by other more adequate sentence.
2. The Authors should justify dose of GnIH used in this work. Were control animals treated with vehicle? Furthermore, what is physiological concentration of GnIH in the circulation and how ist releted to dose used in this study? Furthermore, regulation of GnIH by metabolic status should be also elaborated.
3. Tabele 1. Gene accession no. should be provided.
4. Statistical analysis –ANOVA, type of post hoc test is missing.
5. The Authors found that GnIH-treted mice had reduced expression of insulin while glucagon mRNA expression was enhanced. Since insulin promotes adipogenesis and lipogeneis while glucagon induces lipolysis increased body weight reported in GnIH-treted animals seams to be at least surprising. In my opinion this point should be discussed and commented by the Authors.
6. Listing and discussing study limitation can improve discussion.
7. Figure 2b Y chart. Are results are really shown as mmol/ml? Or maybe ml should be replaced by L? Please clarify this point.
Round 2
Reviewer 1 Report
Please correct "metabolism relative genes" and similar constructions to metabolism-related genes
Language still needs polishing
Author Response
Dear Reviewer 1:
We very much appreciate your careful reading and useful suggestions for our manuscript. We have carefully considered the comments and have revised the manuscript accordingly. Our itemized response list as follows:
1,Please correct "metabolism relative genes" and similar constructions to metabolism-related genes
Response: We followed this suggestion. The manuscript and titles of Y-axis in Fig 3 and Fig 4 were revised.
2,Language still needs polishing
Response: We followed this suggestion. We are getting help from the language editing service of MDPI to proofread the manuscript. We hope these English language improvements will make our paper more acceptable for publication.